# SciRepEval: A Multi-Format Benchmark for Scientific Document Representations

## Abstract

Learned representations of scientific documents can serve as valuable input features for downstream tasks, without the need for further fine-tuning. However, existing benchmarks for evaluating these representations fail to capture the diversity of relevant tasks. In response, we introduce SciRepEval, the first comprehensive benchmark for training and evaluating scientific document representations. It includes 25 challenging and realistic tasks, 11 of which are new, across four formats: classification, regression, ranking and search. We then use the benchmark to study and improve the generalization ability of scientific document representation models. We show how state-of-the-art models struggle to generalize across task formats, and that simple multi-task training fails to improve them. However, a new approach that learns *multiple* embeddings per document, each tailored to a different format, can improve performance. We experiment with task-format-specific control codes and adapters in a multi-task setting and find that they outperform the existing single-embedding state-of-the-art by up to 1.5 points absolute.

## 1 Introduction

Learning representations of whole documents is critical for a variety of NLP tasks including classification, search, and recommendation (Cohan et al., 2020). Recent work has shown how pretrained language models (e.g., (Devlin et al., 2019; Raffel et al., 2020; Brown et al., 2020)) can be tailored to produce high-quality representations of documents with contrastive learning (Xu et al., 2021; Gao et al., 2021; Neelakantan et al., 2022). In the scientific domain, training objectives based on contrastive learning of cross-document links (e.g., citations) have shown further improvements in document-level representation learning (Cohan et al., 2020; Ostendorff et al., 2022b; Mysore et al., 2022). These methods are especially useful because the representations they produce can be indexed and later efficiently consumed by lightweight downstream models without additional fine-tuning.

While there has been significant progress in evaluating generalizability of NLP models (Ye et al., 2021; Sanh et al., 2021), evaluation of scientific document representation models has remained limited. Existing benchmarks either focus on document similarity (Mysore et al., 2021; Voorhees et al., 2021) or include tasks that are highly correlated and not diverse (Cohan et al., 2020).

We introduce SciRepEval, the first benchmark for comprehensive evaluation of document-representation learning models in the scientific domain. Unlike prior work, SciRepEval is large and includes a collection of highly diverse tasks, thus encouraging research on generalization (including instance-level, cross-task and cross-domain generalization). It consists of 25 realistic tasks that reflect practical use cases of scientific document representations across four formats: text classification, regression, proximity-based ranking (e.g., nearest-neighbor), and ad-hoc search. Eleven of these tasks are new contributions. SciRepEval contains standard sets of both training and evaluation datasets to simplify and standardize comparisons between methods evaluated on the benchmark.

We then use this new benchmark to investigate and improve the generalization ability of document representation models. Following recent work (Cohan et al., 2020; Ostendorff et al., 2022b; Mysore et al., 2022) we further pre-fine-tune a transformer language model (SciNCL; Ostendorff et al., 2022b) to produce high-quality representations for downstream tasks. We hypothesize that condensing all relevant information of the document into a single vector representation might not be expressive enough for generalization across a wide range of tasks. Prior work addresses a similar challenge in the context of document similarity by learning multiple finer-grained representations,

each associated with a different *aspect* of a paper (e.g., task, method, results, etc) (Mysore et al., 2022; Ostendorff et al., 2022a). In contrast, we aim to learn effective representations for multiple downstream task *formats*.

Following recent success in multi-task learning in NLP (Ye et al., 2021; Sanh et al., 2021), we explore large-scale multi-task training in the context of scientific document representations, where we apply suitable optimization objectives for the various task formats in SciRepEval. i.e., cross-entropy loss for classification, triplet loss for proximity/ad-hoc search, and mean square error loss for regression. We explore two state-of-the-art techniques for generating format-specific document representations: using control codes (Keskar et al., 2019; Raffel et al., 2020) as input indicating the format, and parameter-efficient adapter methods (Houlsby et al., 2019; Pfeiffer et al., 2021; Stickland & Murray, 2019), in which a separate network module is introduced for every task format.

Our experiments investigate: (*i*) if existing document representation methods have the ability to generalize to a highly diverse set of tasks, (*ii*) if multi-task training on diverse data can improve document representation models, and (*iii*) if task-format-specific representations can improve generalization. Through extensive analysis we find that existing state-of-the-art scientific document representation models such as SPECTER (Cohan et al., 2020) and SciNCL (Ostendorff et al., 2022b) struggle with generalizing to all task types. We interestingly find that simple multi-task training on large set of tasks is *not* able to significantly improve the results. However, we learn that multiple task format-specific representations can substantially improve generalization.

To summarize, our contributions are:

(i) SciRepEval, a new comprehensive benchmark of 25 highly diverse and practical tasks for scientific document representation techniques across four different formats, of which 11 tasks are made available for the first time, and six of the tasks are explicitly designed for training.

(ii) An extensive investigation on the generalization ability of state-of-the-art scientific document representation models.

(iii) A set of new multi-task document representation models that, unlike existing methods, can produce representations tailored to different task formats. The new methods show improved generalization over previous work, outperforming prior methods by up to 1.5 points absolute.

## 2 BACKGROUND

**Representing Scientific Documents**   Earlier work aimed at document embeddings used word vectors (J et al., 2016; Le & Mikolov, 2014; Wu et al., 2018), convolutions (Liu et al., 2017; Zamani et al., 2018), bi-encoder networks (Conneau et al., 2017) and BERT-based methods (Reimers & Gurevych, 2019). Recent works have produced large scale language models pre-trained on scientific corpora (Beltagy et al., 2019; Yasunaga et al., 2022; Trewartha et al., 2022). These tend to perform better than general purpose models on scientific domain tasks, and serve as a foundation for learning dense embeddings of scientific documents. Cohan et al. (2020) and Ostendorff et al. (2022b) fine-tune SciBERT (Beltagy et al., 2019) with a triplet loss that encourages papers citing each other to have similar embeddings, using the title and abstract of research papers as the input.

Both Cohan et al. (2020) and Ostendorff et al. (2022b) are evaluated on the SciDocs benchmark. However, 4 of the 7 tasks in SciDocs are overly-simplistic in that the goal is to distinguish 5 real citations from 20 randomly chosen non-citations (further limitations of SciDocs are discussed in section 3 and Appendix F). Hence, the existing techniques work reasonably well on SciDocs. In contrast, SciRepEval provides a more challenging and diverse set of tasks, for both training and evaluation to help motivate methods for producing scientific document representations that can do well across multiple task formats. As a first step in this direction, we attempt to learn task-specific embeddings of the documents by pre-fine-tuning on multiple objectives simultaneously. Related to our approach, there are techniques in learning multiple embeddings per paper (Ostendorff et al., 2022a; Mysore et al., 2022). These methods are, however, orthogonal to ours in that they generate an embedding per paper "facet", while we focus on learning separate embeddings per task format. In addition, these techniques focus only on finer-grained paper similarity, while our aim is producing general embeddings applicable to a variety of task formats.

**Multi-Task Learning Across Formats**   Multi-task learning (Caruana, 1993) with deep neural networks has been shown to improve performance over single-task training for related objectives (Liu

et al., 2015; 2019b). Though unrelated tasks can lead to negative transfer, recent work has shown that simply increasing the number of tasks tends to yield better performance in multi-task learning (Aghajanyan et al., 2021; Aribandi et al., 2022; Padmakumar et al., 2022). Aghajanyan et al. (2021) pre-fine-tune pre-trained language models simulataneously on 46 tasks across 4 task types before fine-tuning on the downstream task. Aribandi et al. (2022) pre-train T5 (Raffel et al., 2020) on a combination of C4 span denoising and 107 other tasks across 8 task families. Ye et al. (2021) introduce an ontology of 160 tasks for few shot multi-task training. Unlike these task families, which are divided primarily by semantics (e.g., classifying sentiment vs classifying entailment), the training tasks in SciRepEval consist of 8 large-scale scientific datasets across the four task *formats*. Since our goal is to evaluate final document representations, rather than fine-tune on individual downstream tasks like the above approaches, we follow SPECTER (Cohan et al., 2020) and directly apply the representations as features to the tasks.

**Adapters for Multiple Tasks** Adapters were introduced by Houlsby et al. (2019) for parameter efficient training of transformers (Vaswani et al., 2017). A small number of trainable parameters are added to each layer, while freezing the base encoder. This strategy is similar to that of ELMo (Peters et al., 2018), which learned task-specific weightings for the biLSTM layers. To apply adapters to multi-task learning, Pfeiffer et al. (2021) define a two-step process they call Fusion. First, individual adapter modules are trained for every task. The second step introduces task-specific fusion modules at each layer which attend to (i.e. fuse) all the previously pre-fine-tuned adapters, keeping them fixed. Similarly, Stickland & Murray (2019) introduced Projected Attention Layers (PALs) with adapters and self-attention modules for every task, but the entire network is trained simultaneously.

**Control Codes** Control codes can be defined as token(s) pre-pended to the input to serve as additional signals to the model. Keskar et al. (2019) use control codes as prompts to govern style, content, and task-specific behavior for conditional text generation. Tay et al. (2022) use control codes to switch between three de-noising modes during pre-training, and associate a downstream task with a particular mode during fine-tuning. Zhang et al. (2022) apply control codes in the context of dense retrieval to produce multiple representations covering different aspects of the same document, allowing them to match queries written from multiple perspectives. In contrast to this past work, we use control codes to indicate target task format for the embedding output by the model, and demonstrate how this is effective for producing paper embeddings across different formats.

## 3 SCIREPEVAL

We introduce SciRepEval, a benchmark suite of 25 tasks across four formats for training and evaluating multi-task embeddings of scholarly papers. SciRepEval aims to enable comprehensive evaluation of paper embeddings by providing (1) a highly diverse set of tasks—spanning multiple task formats such as classification, regression, proximity and ad-hoc search—to challenge the general-purpose applicability of embeddings, (2) realistic tasks that reflect practical use cases of paper embeddings, and (3) a standard set of both training and evaluation datasets to simplify comparisons between methods evaluated on the benchmark.

The previous scholarly paper embedding benchmark is SciDocs (Cohan et al., 2020), which includes two classification tasks, four nearest neighbors tasks, and one recommendation task. SciRepEval includes SciDocs as a subset, but addresses several key limitations:

 (i) The four nearest neighbor tasks in SciDocs are constructed to distinguish a related document from randomly selected negatives given a query document, which might be too easy and not representative of real tasks in scholarly information retrieval. SciRepEval has more realistic tasks such as search, author disambiguation, and paper-reviewer matching among others.

 (ii) For the methods evaluated in Section 5, we found that the SciDocs recommendations task was noisy and had limited power to distinguish between different embeddings. The test set includes only 1000 clickthrough events, and the use of propensity weighting means that an even smaller number of examples dominate test set performance. While SciRepEval includes SciDocs as a subset, we exclude the recommendations task.

 (iii) The tasks in SciDocs were constructed to be used *only* for evaluation, and have few-enough samples that training on SciDocs is impractical (see Table 1). In SciRepEval, eight of the largest tasks across the four formats are used for training, while the rest out-of-train tasks are reserved for evaluation. This enables the study of multi-task approaches, rather than relying

Table 1: Summary of the SciRepEval benchmark tasks across the four formats - classification (CLF), regression (RGN), proximity (PRX) and adhoc search (SRCH). The models in section 6 are first trained on the in-train tasks and then benchmarked on their held-out sets as well as the 17 test tasks. Information retrieval tasks have **Q** queries with **P** candidate pairs and the S2AND task has **X** clusters with **Y** author-paper pairs. **S**: Silver, **G**: Gold. SciDocs is evaluated as per Cohan et al. (2020).

| Task Format | Name | Train + Dev | Test | Eval Metric | Source |
|---|---|---|---|---|---|
| | | *In-Train* | | | |
| CLF | MeSH Descriptors | 2,328,179 | 258,687 | Macro F1 | **This work** |
| | Fields of study (FoS) | 676,524 S | 471 G | Macro F1 | **This work** |
| RGN | Citation count | 202,774 | 30,058 | Kendall's $\mathcal{T}$ | **This work** |
| | Year of Publication | 218,864 | 30,000 | Kendall's $\mathcal{T}$ | **This work** |
| PRX | Same Author Detection | **Q:** 76,489 **P:** 673,170 | **Q:** 13,585 **P:** 123,430 | MAP | Subramanian et al. (2021) |
| | Highly Influential Citations | **Q:** 65,982 **P:** 2,004,688 | **Q:** 1,199 **P:** 54,255 | MAP | **This work** |
| | Citation Prediction Triplets | 819,836 | — | *not used for eval | Cohan et al. (2020) |
| SRCH | Search | **Q:** 723,343 **P:** 7,233,430 | **Q:** 2,585 **P:** 25,850 | nDGC | **This work** |
| | | *Out-of-Train* | | | |
| CLF | Biomimicry | — | 11,057 | Binary F1 | Shyam et al. (2019) |
| | DRSM | — | 7,520 S; 955 G | Macro F1 | Burns (2022) |
| RGN | Peer Review Score | — | 10,210 | Kendall's $\mathcal{T}$ | **This work** |
| | h-Index of Authors | — | 8,438 | Kendall's $\mathcal{T}$ | **This work** |
| | Tweet Mentions | — | 25,655 | Kendall's $\mathcal{T}$ | Jain & Singh (2021) |
| PRX | S2AND | — | **X:** 68,968 **Y:** 10,942 | $B^3$ F1 | Subramanian et al. (2021) Mimno & McCallum (2007) |
| | Paper-Reviewer Matching | — | **Q:**107 **P:** 1,729 | P@5, P@10 | Liu et al. (2014) Zhao et al. (2022) |
| | Feeds-1 | — | **Q:** 423 **P:** 4,223 | MAP | **This work** |
| | Feeds-M | — | **Q:** 9025 **P:** 87,528 | MAP | **This work** |
| SRCH | Feeds Title | — | **Q:** 424 **P:** 4,233 | MAP | **This work** |
| | TREC-CoVID | — | **Q:** 50 **P:** 69,318 | nDCG | Voorhees et al. (2021) |
| | | *SciDocs* | | | |
| CLF | MAG | — | 23,540 | Macro F1 | Cohan et al. (2020) |
| | MeSH Diseases | — | 25,003 | Macro F1 | Cohan et al. (2020) |
| PRX | Co-view | — | **Q:** 1,000 **P:** 29,978 | MAP, nDCG | Cohan et al. (2020) |
| | Co-read | — | **Q:** 1,000 **P:** 29,977 | MAP, nDCG | Cohan et al. (2020) |
| | Cite | — | **Q:** 1,000 **P:** 29,928 | MAP, nDCG | Cohan et al. (2020) |
| | Co-cite | — | **Q:** 1,000 **P:** 29,949 | MAP, nDCG | Cohan et al. (2020) |

solely on the citation signal. The training data in SciRepEval also has a large scale representation in multiple domains as discussed in Appendix D.

(iv) Four of the tasks in SciDocs have very high model-performance correlations between them (greater than 0.99), indicating that the diversity of the tasks is limited. See Appendix F for more details.

The tasks in SciRepEval are summarized in Table 1. They are a mixture of existing and new datasets. Datasets with at least 100,000 instances (triplets for proximity/ad-hoc search) are *in-train* datasets used for training while others are *out-of-train* used only for evaluation. Although SciDocs tasks are used as out-of-training evaluation tasks, we report their performance in a separate category.

Next, we briefly describe each of the task formats and their component tasks. Full details are provided in Appendix A. Except for *Search*, all the tasks use paper embeddings created from a combination of paper title and abstract as the input. Search requires additional metadata (subsection 4.1) which is concatenated to the title and abstract before producing the paper representation.

**Ad-Hoc Search** In ad-hoc search tasks, we are given a textual query and the task is to rank a set of candidate papers by relatedness to the query. Ad-hoc search is a critical mechanism for paper discovery in practice, and we gather multiple real-world data sets for training and evaluation. One evaluation dataset comes from previous work, TREC-CoVID (Voorhees et al., 2021), a biomedical challenge task that ranks papers from CORD-19 (Wang et al., 2020b) in response to textual search queries. Two other datasets are newly introduced in our work: a 'feeds' dataset taken from a scholarly paper recommendation system, where we treat the user-specified feed name as the topic query, and the goal is to rank the papers the user has annotated as relevant to the feed above those annotated as irrelevant. Finally, for training, we release a new large data set of more than 700,000 clickthrough events from a scholarly search engine which we term as the *Search* task.

To evaluate an embedding set on ad-hoc search, we rank candidate papers by increasing Euclidean distance between the query embedding and the candidate paper embeddings. `Pytrec_eval` (Van Gysel & de Rijke, 2018) is used to calculate the ranking metrics. Normalized Discounted Cumulative Gain (nDCG) is used for Search and TREC-CoVID tasks as the true relevance score can be $> 1$. For the feeds tasks which have binary labels, we use Mean Average Precision (MAP).

**Proximity** Similar to ad-hoc search, proximity tasks involve ranking a set of candidate papers by their relatedness to a query, except the query in this case is not textual but instead a paper. Proximity-based tasks form a basis for paper-based retrieval and recommendation, and for estimating paper similarity for use in applications like author disambiguation. We include a total of eleven proximity-based tasks, including four evaluation tasks from SciDocs (predicting citations and co-citations, and predicting co-viewed or co-read papers), and two others from previous work: the S2AND author disambiguation task (Subramanian et al., 2021) with paper similarity features, and Paper-Reviewer Matching, where candidate reviewers are ranked by expert annotators based on the similarity of their papers to the query paper to be reviewed. The Paper-Reviewer Matching task combines three existing datasets (Mimno & McCallum, 2007; Liu et al., 2014; Zhao et al., 2022) which we describe in more detail in subsection A.2. We also introduce five new proximity tasks including two feeds evaluation tasks from the recommender discussed above, where one or multiple relevant papers serve as queries. For training, we include three large-scale datasets aimed at predicting same-authors, citations (via triplets) (Cohan et al., 2020), and influential citations, which we define as four or more citations of the same paper in the text of a single paper.

For evaluating embeddings in proximity tasks, we rank candidates by Euclidean embedding distance, with MAP as the evaluation metric except for S2AND, which uses $B^3$ F1 (Bagga & Baldwin, 1998), and Peer Review Matching, which uses precision@5 and @10.

**Classification** Paper classification, in which the input is a paper and the output is a topical category, is a foundational task for document organization and discovery. Apart from the two SciDocs classification tasks (MAG and MeSH Diseases), we take four additional classification tasks, including a binary task to predict whether a paper is relevant to biomimicry (Shyam et al., 2019), two biomedical classification tasks, namely DRSM from Burns (2022) and MeSH Descriptors classification (Lipscomb, 2000), and a new large-scale field of study (FoS) multi-label training set of more than 500K papers with silver FoS labels based on publication venue.

We evaluate embeddings on classification by scoring their performance as features within linear support vector classifiers. Results for these tasks are evaluated using F1 score (which may be micro- or macro-F1 depending on the dataset, indicated in Table 1). To better understand how embeddings perform in data-scarce regimes, we also construct two few-shot versions each from both out-of-train classification datasets and the FoS dataset subset for which we have manually annotated gold labels.

**Regression** We also consider a set of regression tasks where the goal is to predict a continuous quantity for a given paper. For evaluation, we consider predicting three numeric attributes related to prominence or quality: Tweet Mentions (Jain & Singh, 2021), and the peer review rating and maximum h-index of authors for a collection of ICLR papers obtained from OpenReview[1](forming two new datasets). For training, we introduce two additional datasets of more than 200K examples each, predicting citation count and year of publication.

We evaluate embeddings on regression tasks by scoring their performance when used as features within linear support vector regression models. Results for these tasks are evaluated using the Kendall's $\tau$ rank correlation between the true and predicted labels.[2]

## 4 MULTI-FORMAT REPRESENTATION LEARNING

Typical approaches for learning document embeddings produce a single embedding for every task (Cohan et al., 2020; Ostendorff et al., 2022b). We hypothesize that a single embedding will be insufficient for generalizing across a diversity of downstream tasks when the embeddings are used as features in lightweight classifiers. At the other extreme, learning embeddings for each task sepa-

---

[1]https://api.openreview.net

[2]We found in our experiments that Pearson's $\rho$ and Kendall's $\tau$ produced similar relative results between models. We did not use MSE because its values are unbounded and could skew the overall average across the datasets in the benchmark.

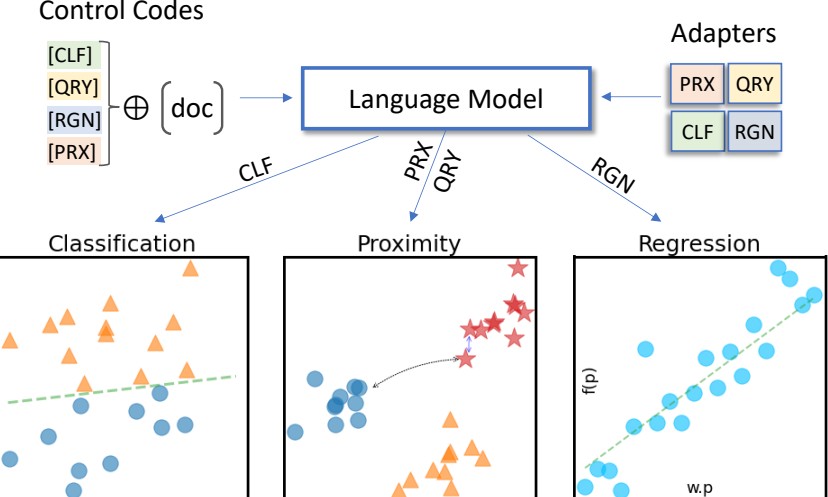

Figure 1: Generating multi-format embeddings. A task format is either associated with a task-specific control code supplied with input document, or adapter blocks attached to the model.

rately does not allow generalization to new tasks and also incurs significant storage costs scaling with the number of tasks. We propose method for learning a distinct document embedding for each task format, using a multi-task learning framework.

We assume we are given labeled data from a set of tasks for our four formats (ad-hoc search, proximity, classification, and regression), and we learn models capable of producing an embedding for any given (paper, format) pair. Here, papers are represented in terms of their title and abstract. Our goal is for these embeddings to be used in lightweight classifiers/regressors as well as in nearest neighbor tasks, which we evaluate both on held-out data from the training tasks, *and* on new held-out tasks.

To help build intuition for why different embedding sets for different task formats may be helpful, Figure 1 illustrates the qualitative distinctions between the task formats. In general, an embedding space performing well for one task format may be less suited to the others; for example, the classification space provides an error-free linear classifier, but its nearest neighbor pairs are not always of the same class. Empirically, we find that learning specialized embeddings per format improves performance, and that embeddings trained on a format tend to perform better on held-out tasks with the same format (see Table 4). Further, partitioning randomly (as discussed in Section 7) was less effective than the format-based partitioning. Nonetheless, format-based partitioning is just one choice of many and experimenting with other partitioning schemes is an important item of future work.

## 4.1 MODEL

We follow Cohan et al. (2020) in using a pretrained transformer encoder as our base model. A scientific document is given as input to the encoder as a concatenation of its title and abstract separated by the [SEP] token [3]. Unlike Cohan et al. (2020), we use three different types of training objectives suitable for each format to train the model as described in subsection 4.2. We explore two methods to learn separate embeddings for each task form: control codes and adapters as shown in Figure 1.

**Control Codes** In the control code approach, we prepend a special per-format token (see Table 6 in the appendix) to the input and pass it to the transformer model, taking the final layer representation corresponding to this token as the document embedding and feeding it as input to the task-specific head (described in Section 4.2).

**Adapters** We also experiment with adapters which have been shown to be effective for multi-task learning. In particular, we explore Adapter Fusion (Pfeiffer et al., 2021) and PALs (Stickland & Murray, 2019) methods, each of which introduces task-specific adapters and attention modules at every transformer layer. Since our goal is to learn different embeddings for different task formats,

---

[3]For the Search task, additional metadata like paper venue and year of publishing is also supplied.

we create modules for each task format rather than each task, and the final hidden representation of the [CLS] token output via the adapter is taken as the corresponding embedding of the document.

## 4.2 TRAINING

We train the model in a multi-task setup with task-heterogeneous batching (Aghajanyan et al., 2021). For classification and regression tasks, we use a linear head atop the base transformer encoder[4]. We train on both multi-class and multi-label tasks, using Cross Entropy loss for the former and Binary Cross Entropy (BCE) with sigmoid activation for the latter. For regression we minimize the Mean Square Error (MSE) loss.

For proximity and ad-hoc search tasks we use the triplet loss as in Cohan et al. (2020). For these task forms, given a query, a relevance score accompanies each candidate. The query can be a document (for which we wish to find similar documents) or a raw textual query. Each training instance in this setup is a triplet consisting of a paper or plain text query $\mathcal{Q}$, a positive candidate paper $\mathcal{P}+$ and a negative candidate $\mathcal{P}-$, where $\mathcal{P}+$ has a higher score than $\mathcal{P}-$. Then, we optimize the triplet loss:

$$L_{triplet} = \max\{d(\mathcal{Q}_E, \mathcal{P}_E^+) - d(\mathcal{Q}_E, \mathcal{P}_E^-) + \epsilon, 0\} \qquad (1)$$

where $d$ is the Euclidean distance used as a measure of similarity between the query embedding $\mathcal{Q}_E$ and candidate embeddings $\mathcal{P}_E^+$ and $\mathcal{P}_E^-$, and $\epsilon$ is the margin hyperparameter whose value is chosen as 1 based on preliminary experiments.

## 5 EXPERIMENT SETUP

**Training Data** We train our multi-format models on the 8 large scale in-train tasks detailed in Table 1. For the proximity and ad-hoc search tasks, we create up to 5 examples for each query by sampling positive and negative papers from its candidate pool. We limit the number of training samples from each task to at most 600K.[5] The resultant training and validation data sets consist of a total of 3.27M and 446K instances respectively.

**Transformer Baselines** As a first step, we evaluate the existing document representation methods on our benchmark. These include the transformer encoders SciBERT (Beltagy et al., 2019) – a language model pre-trained on scientific corpora; and SPECTER (Cohan et al., 2020), SciNCL (Ostendorff et al., 2022b) and ASPIRE (Mysore et al., 2022). ASPIRE produces representations for aspect-based matching between query and candidate papers which is a similar setting as our proximity tasks. Hence we only evaluate it on that specific subset and report the results in Appendix C. Next, for our multi-format baselines, we initialize with SciNCL which is the state of the art on Sci-Docs, and then further train it in a multi-task setup on the in-train tasks both with (MTL CTRL) and without the control codes (MTL CLS). Finally, to compare the control codes-based approach with the adapter techniques, we experiment with the BERT PALs and Fusion architectures, keeping SciNCL as the base model in both. Fusion being a two step process, first introduces task format specific adapters (Adapters) and then the fusion modules (Adapter Fusion). The MTL CTRL and adapter approaches produce multiple representations per document while MTL CLS produces a single representation similar to existing methods. We use the PyTorch implementations of the models by HuggingFace[6]. The specific training configurations are described in Appendix B.

## 6 RESULTS

Table 2 shows the evaluation of all our transformer baselines producing both single and multiple representations per document on SciRepEval. Our benchmark includes diverse tasks with a variety of different evaluation metrics, and following previous work (e.g., Wang et al., 2019) we report an average of the individual task metrics (which each range from 0-100). The pre-fine-tuned multi-format variants outperform the vanilla models on average, and we also find that all the approaches that produce multiple representation types outperform, by up to 1.5 points, the MTL CLS model, which learns only a single representation shared for all tasks. The adapter variants are better than MTL CTRL overall, and result in an improvement of 0.6-1.3 points on the out-of-train tasks with task-format specific adapters performing the best.

---

[4]The linear heads are thrown away after training.

[5]Performance with smaller dataset samples - max 400K samples/tasks was relatively poor.

[6]https://huggingface.co/models

Table 2: Evaluation results on SciRepEval in multiple settings. MTL CLS generates a single embedding for all tasks, MTL CTRL (control codes) and Adapter variants (Adapters, PALs, and Adapter Fusion) produce an embedding per task format. We consider an ensemble approach that averages the MTL CTRL and Adapter embeddings. For models we trained, we report the mean and standard deviation (in parentheses) across 5 runs with different seeds. The best results are highlighted in **bold**. We conduct one way analysis of variance (ANOVA) with Tukey's test (Haynes, 2013) for $\alpha = 0.05$ across multiple settings and underline those not statistically significantly different from the best.

| Model | In-Train | Out-of-Train | SciDocs | Average |
|---|---|---|---|---|
| SciBERT | 51.5 | 52.5 | 69.0 | 58.2 |
| SPECTER | 54.7 | 57.4 | 89.1 | 68.0 |
| SciNCL | 55.6 | 57.8 | **90.8** | 69.0 |
| SciNCL + MTL CLS | 60.0 (0.11) | 57.0 (0.28) | 89.5 (0.08) | 69.4 (0.12) |
| SciNCL + MTL CTRL | 61.9 (0.22) | 57.8 (0.4) | 89.7 (0.16) | 70.2 (0.23) |
| SciNCL + Adapters | 61.9 (0.11) | __59.1__ (0.06) | 90.3 (0.07) | __70.9__ (0.03) |
| SciNCL + PALs | 61.9 (0.26) | 58.4 (0.28) | 90.1 (0.09) | 70.6 (0.11) |
| SciNCL + Adapter Fusion | 62.0 (0.04) | __58.8__ (0.26) | 90.1 (0.11) | 70.8 (0.14) |
| SciNCL + Adapters + MTL CTRL | **62.5** (0.01) | **59.1** (0.15) | __90.6__ (0.01) | **71.2** (0.03) |

Table 3: Results for multi-format training with SciBERT and SPECTER as base models. For brevity, we report only the single adapters results due to their additional advantage of computation efficiency. The best results for each base model are underlined.

| Model | In-Train | Out-of-Train | SciDocs | Average |
|---|---|---|---|---|
| SciBERT + MTL CLS | 59.4 | 57.1 | 88.9 | 69.0 |
| SciBERT + MTL CTRL | 62.5 | 57.8 | 89.6 | 70.4 |
| SciBERT + Adapters | 62.6 | 58.0 | 90.1 | 70.6 |
| SciBERT + Adapters + MTL CTRL | __62.9__ | __58.6__ | __90.4__ | __71.1__ |
| SPECTER + MTL CLS | 60.0 | 56.5 | 89.0 | 69.0 |
| SPECTER + MTL CTRL | 61.9 | 58.0 | 89.4 | 70.2 |
| SPECTER + Adapters | 61.5 | 58.9 | 89.6 | 70.5 |
| SPECTER + Adapters + MTL CTRL | __62.2__ | __59.1__ | __89.9__ | __70.9__ |

Further, as shown in Table 5, the control codes and adapters are the most efficient in terms of model size and computational efficiency. Hence, we try to improve upon each by combining representations from the Adapter model and the MTL CTRL model by averaging them[7], and we find that these combined embeddings outperform the individual ones consistently across the in-train, out-of-train, and SciDocs settings. All the models except SciBERT (not pre-trained with a citation objective) perform well on SciDocs, with vanilla SciNCL being the best. ASPIRE, as reported in Appendix C, performs well on SciDocs but not on other similar tasks in SciRepEval.

**Alternative Base Models**   To confirm that our findings hold across multiple base models, we compare MTL CLS, MTL CTRL and adapters with SPECTER and SciBERT as the base models. Table 3 shows that the MTL CTRL token and the adapters approaches still substantially outperform the MTL CLS approach, suggesting that the efficacy of using an embedding per task format instead of a single embedding per document holds across a range of base model types.

## 7   ANALYSES

**Specialization of Control Code Embeddings**   Our hypothesis is that by training embedding spaces on particular task formats, they will become more accurate for tasks of that format than for others. We test this hypothesis by sampling one in-train and one out-of-train[8] task of every format (for ease of computation) and applying *all* the control codes to them for evaluation. As shown

---

[7]We also tried concatenating the embeddings in preliminary experiments, which yielded similar results but doubled the embedding size.

[8]In-train: FoS, Citation Count, Same Author Detection, Search; Out-of-train: DRSM, Peer Review Score, Peer-Reviewer Matching, TREC-CoVID

Table 4: Cross task analysis for control codes. The best results for each task format across all control codes is underlined. These are represented in the diagonal for both in-train and out-of-train tasks suggesting that format based partitioning in multi-task training produces effective document representations suitable for the corresponding format.

| Task format | Control Code Embeddings Used | | | | | | | |
| --- | --- | --- | --- | --- | --- | --- | --- | --- |
| | *Out-of-Train* | | | | *In-Train* | | | |
| | CLF | RGN | PRX | QRY | CLF | RGN | PRX | QRY |
| Classification | 66.9 | 63.5 | 65.0 | 64.5 | 38.7 | 32.9 | 31.9 | 31.0 |
| Regression | 16.2 | 20.4 | 18.1 | 17.5 | 29.7 | 46.8 | 43.7 | 42.9 |
| Proximity | 43.8 | 43.1 | 44.7 | 44.7 | 87.1 | 82.2 | 89.0 | 88.3 |
| Ad-hoc search | 86.7 | 85.2 | 85.3 | 90.5 | 73.9 | 75.6 | 77.5 | 78.5 |

Table 5: Parameter and (relative) runtime efficiency comparison of models. MTL CTRL and Adapters are similar in runtime, but the PALs and Fusion variants add significant computation costs.

| Model | Parameters per Task Form | Training Time | Inference Time |
| --- | --- | --- | --- |
| MTL CTRL | 768 | 1x | 1x |
| PALs | 2M | 1.42x | 1.29x |
| Adapters | 1M | 0.96x | 1.05x |
| Adapter Fusion | 22M | 1.32x | 1.69x |

in Table 4, the control codes trained on a task format perform best for tasks of that format, for both in-train and out-of-train.

As an extension to this experiment we also analyze how well the control code representations work when the encoder is trained on tasks which are randomly grouped together as opposed to by task format. We take the mean evaluation metrics produced from 5 random partition runs. On the out-of-train tasks, the corresponding control codes for classification, regression, proximity and ad-hoc search show a gain of +0.2, +3.9, +4.5 and +2.2 points respectively over random partitioning. Similarly, for in-train tasks the control codes are better by +5.2, +3.8, +1.2 and +1.3 points respectively. The results suggest that representations specific to each task format do lead to better results overall.

Finally, to study training affinity among the task formats, we pre-fine-tune on a maximum of two formats at once. Appendix G reveals that combined multi-task training on similar task formats like regression/classification and proximity/adhoc-search results in performance gains, but only on related tasks. Training on all the tasks yields better results on average across the task formats.

**Efficiency** While the variants producing representations based on task-format serve as strong baselines on the SciRepEval benchmark as shown in Table 2, efficiency is another important consideration in practice. As shown in Table 5, the control code approach only requires one new control code embedding per format, and does not affect training time. PALs, by contrast, introduces new attention layers and trains the entire network, increasing training time, and Adapters adds and only trains half as many parameters as PALs. Fusion layers introduce 10x as many parameters as PALs leading to 2x more time on inference. Training and inference times are measured on runs with 1k and 10k samples, respectively.

## 8 CONCLUSION

We introduce SciRepEval, a benchmark for scientific document representation methods with 25 tasks across four task formats. On this benchmark, we show that learning a separate document representation for each task format substantially improves task performance compared to learning a single representation for all tasks. Future work could address limitations of our work by evaluating partitioning schemes beyond task format, crafting higher-fidelity metrics to account for the diversity of tasks in SciRepEval (which vary in sensitivity and in relevance to downstream applications), or further exploring how accuracy varies with computational and storage cost.

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

# A  SCIREPEVAL TASKS

## A.1  AD-HOC SEARCH

**Search**  We used clickthrough data from an academic search engine. Only search queries with at least 10 results were included, and a set of heuristic rules were applied to exclude likely noise and bots. We removed author queries when the query was judged to contain any person tokens by named entity recognition (Honnibal et al., 2020).

**Feeds**  Research feeds help researchers maintain a library of papers they are currently reading and also index them by topics. We use anonymized research feeds data from an academic search engine that recommends papers based on a user's library. This data includes information on whether users found the recommendations relevant or not. The data contains 430 feeds which have more than five positive and two negative paper annotations from users. We use this to create the Feeds-1, Feeds-M and Feeds Title tasks. The first two are proximity tasks and are described in section 3.

*Feeds Title:* The title of a research feed provided by the user usually indicates the topic of scientific articles in the feed. While the other two datasets have papers as query and belong to the proximity family, this dataset is classified as ad-hoc search, as the query is a short text snippet rather than a paper. We remove feeds with generic titles like 'My field' and 'Final Project'; replace abbreviations with their long forms where possible and filter out feeds with non-English titles.

**TREC-COVID**  TREC-COVID was introduced by Voorhees et al. (2021) as a biomedical literature search task relevant to COVID-19. The dataset consists of 50 search queries and candidate literature from the CORD-19 corpus (Wang et al., 2020b) along with their relevance scores on a scale of 0-2. Each query consists of a short title, a question asking the required information and a narrative describes briefly exactly the type of information that the results should have. For our evaluation we combine these fields into a single text separated by the [SEP] token.

## A.2  PROXIMITY

**S2AND and Same Author Detection**  The S2AND dataset (Subramanian et al., 2021) contains signatures (author-paper pairs) that are clustered according to which author mentions refer to the same person. Due to the high resource requirements of running the original S2AND evaluation, we create S2AND-mini, a version of S2AND with only 1000 blocks from each of S2AND's dataset sources and at most 500 signatures per block. Our evaluation of S2AND-mini follows the original evaluation of S2AND; that is, our method's document embeddings are used along with author and paper metadata to create features for a clustering algorithm that consists of a pairwise scoring model followed by greedy agglomerative clustering. We use $B^3$ F1 (Bagga & Baldwin, 1998) as in the original paper for evaluation.

We also use S2AND to create the data for our same-author detection task. Unlike the original S2AND evaluation, our same-author task uses only paper embeddings without any additional author or paper metadata, which allows us to directly train the embedding model on the data. Same-author

detection is formulated as a triplet ranking task; given three papers of which two share an author, the goal is to find the matching pair.

**Feeds-1** We re-purpose the feeds dataset from section A.1 for this and the next task. The first paper added to a feed chronologically serves as the query. The next 5 positive user annotations are considered relevant and 5 negative candidates are sampled either from user annotations or randomly.

**Feeds-M** Given $K$ positive papers annotated in a feed (assuming $K > 5$), we use the first $M = K - 5$ as queries. For every query, the positive candidates are sampled from all the papers the user positively annotated after the query paper was added to their feed, and negative candidates are sampled from user annotations or randomly.

**Peer Reviewer Matching** In this task the goal is to judge whether a given paper is relevant to a potential reviewer. As data for this task is hard to obtain at scale due to the double-blind nature of many conferences and journals, we combine multiple existing reviewer-paper matching datasets:

- Mimno & McCallum (2007), with 393 paper-review relevance ratings from a corpus of 148 NeurIPS 2006 papers and 364 reviewers, annotated by nine human experts.

- Liu et al. (2014), an extension of Mimno & McCallum (2007) which adds 766 additional paper-review annotations.

- Zhao et al. (2022), with 694 paper-reviewer relevance ratings from a corpus of 75 papers and 1833 reviewers from the IEEE ICIP 2016 conference, annotated by 3 human experts.

All datasets have been annotated on the same 0-3 relevance rating scale. The candidate reviewers are all researchers, and we embed all the papers written by them using our models. To obtain the model's score for each candidate reviewer, we compute the cosine similarity between the query paper and each of the candidate's papers, and take the mean of the top 3 similarities as the score. We consider two ways to map the 0-3 relevance judgements to binary labels—hard and soft decision—where for the soft decision a score of 2 or 3 is considered relevant and for hard decision only a score of 3 is considered relevant. Precision at 5 (P@5) and 10 (P@10) results are used as the final metric, which ultimately results in four numbers (P@5 and P@10 for each of hard and soft decisions), which are averaged to produce the single number reported in our final results for this task.

**Highly Influential Citations** In this task, given a paper $A$ and paper $B$, we aim to predict whether $B$ is highly influenced by $A$. As measuring influence is subjective and human annotation is expensive, we approximate influence by counting the number of times $A$ is cited in the text of $B$. If $A$ is cited at least 4 times, we consider it to be highly influential (a positive example in our triplet-based loss); otherwise, we consider it to be a negative example. During evaluation, we sample query papers which have at least 5 positive candidates and compute the L2 distance for similarity ranking. Note that our definition of 'influential' differs from that in Valenzuela et al. (2015).

**Citation Prediction (SPECTER Pre-training Triplets)** This is the task and dataset used for pre-training in Cohan et al. (2020). It is based on citation links between scientific documents where each instance is a triplet consisting of a query, a positive and a negative paper. Each query can have up to five triplets, where the positives are sampled from papers directly cited by the query and negatives are chosen either randomly (easy) or from citations of citations (hard). 3 easy and 2 hard difficult are chosen for each query. To evaluate the effectiveness of this pre-training we follow Cohan et al. (2020) and use SciDocs for evaluation, excluding the recommendations task.

### A.3 CLASSIFICATION

**MeSH Descriptors** Medical Subject Headings (MeSH) (Lipscomb, 2000) indexes biomedical publications into a categorical hierarchy consisting of descriptors which refer to topic headings and specific aspect related to a topic respectively. The dataset is a collection of scientific documents belonging to the 30 most frequently occurring top level MeSH descriptors and having exactly one qualifier. We filter out the records that don't have an associated qualifier. The descriptors thus serve as the labels in the multi-class classification task.

**Fields of Study (FoS)**    The FoS task is a multi-label classification problem where each scientific document is assigned one or more classes out of 23 possible fields. For gold test data, we manually labeled 471 papers into at most three fields-of-study. For silver training data, we assumed that a paper within a venue generally falls within a narrow set of fields and manually assigned FoS labels to publication venues. We then propagated the venue labels to the papers published therein.

To evaluate different data sizes, we obtain the F1 score on the gold data in three settings: 5-shot, 10-shot, and the complete gold test set. The average of these scores is treated as the score for this task when computing the overall average score for the benchmark.

**Disease Research State Model (DRSM)**    DRSM (Burns, 2022) is a collection of Pubmed papers that deal with six specific aspects of rare diseases. The gold data is annotated by in-house experts and used for evaluation, while the silver data is generated by annotation service providers with medical expertise.

Similar to FoS, we obtain the F1 score on 24-shot, 64-shot, and full data, then average the results before computing the final benchmark score.

**Biomimicry**    We sample tags for a set of papers in the PeTaL database (Shyam et al., 2019) to create a binary classification dataset with labels indicating whether each paper is about biomimicry. The data is unbalanced, with only 13% positive samples. We evaluate 16-shot, 64-shot, and full-data setup and take the mean to get the final score.

### A.4    REGRESSION

**Citation Count**    We sample a collection of scientific articles published in 2016 from the set of papers in the search dataset described in A.1, so that a 5 year period has passed for them to collect citations. Each article has at least one citation, and the citation counts are converted to log scale.

**Year of Publication**    The aim of this task is to determine research trends by predicting the year of publication of a scientific article. We sample publications from the search dataset with a publication date after the year 2005 and scale the years so that their values are between 0 and 1. Further, since this task is used for training along with citation count prediction, and to align the loss scales, the labels are scaled by the mean of the labels in citation count for parity.

**Peer Review Score**    We use the OpenReview API[9] to collect paper metadata and corresponding review scores for ICLR conferences from 2017 to 2022. Each reviewer in ICLR assigns a final rating in the range [0-10], and we take the mean rating as the label for every paper.

**h-Index of Authors**    In this task the goal is to predict the maximum h-Index of any of the authors of a scientific publication. We re-use the peer review score dataset, obtain the h-Index of all the authors for each paper using the Semantic Scholar API[10], and pick the max as the label. The labels are normalized to lie between [0,1].

**Tweet Mentions**    The goal of this task is to predict the combined number of a paper's mentions and retweets. We post-process the dataset created by Jain & Singh (2021) containing tweets about Arxiv papers between 2010-19. The sum of normalized counts of mentions and retweets is finally considered as the score to be predicted.

## B    IMPLEMENTATION DETAILS

During pre-training, all the tasks with the same format share their task-format specific parameters. The control code based paradigm introduces four new (randomly-initialized) special tokens to the vocabulary. We try initializing these additional parameters randomly, with the [CLS] token and a combination of [CLS] with some noise. However, it has little impact on the resulting model

---

[9]https://api.openreview.net
[10]https://api.semanticscholar.org/

Table 6: Assigned input formats and control codes for each task form. `[CLF]`, `[RGN]`, `[PRX]` and `[QRY]` are special tokens, `doc` is the input.

| Task form | Input format |
|---|---|
| Classification | `concat([CLF],doc)` |
| Regression | `concat([RGN],doc)` |
| Proximity | `concat([PRX],doc)` |
| Ad-hoc Search | `concat([QRY]/[PRX],query/doc)` |

performance with random initialization being better on average. Further, we also tried loss weighting strategies (Chen et al., 2018; Liu et al., 2019a) but our preliminary experiments produced better results without any scaling so we didn't explore it further. All the base models are trained for two epochs on two 48GB NVIDIA Quadro RTX 8000 GPUs with 16 bit precision, an effective batch size of 256, and a maximum input length of 512 tokens. Each batch is sampled with an equal number of examples from each task.[11] We use AdamW (Loshchilov & Hutter, 2019) with $\epsilon = $ 1e-8. The learning rate follows an inverse square root schedule with a linear warmup of 700 steps and peak of 5e-5.

The adapter approaches follow the two step training process and learning rate configurations described in Pfeiffer et al. (2021). One adapter per task family is attached to the base model in both single adapter and fusion stages and is trained for a maximum of 6 and 4 epochs respectively. For PALs one layer is added per task format and the entire network is trained for 2 epochs as in Stickland & Murray (2019).

### B.1 EVALUATION

For classification and regression, we train a linear SVM on each downstream task using the embeddings as input, and we tune the regularization parameter $\mathcal{C}$ via grid search. Multi-class and multi-label classification are configured under the one vs all classifier setting.

## C ASPIRE EVALUATION

Table 7: Comparison of the our SciNCL multi-format methods with ASPIRE on proximity tasks. The best results for each base model are underlined. TS: Text Supervision, OT: Optimal Transport

| Model | In-Train | Out-of-Train | SciDocs | Average |
|---|---|---|---|---|
| TS ASPIRE$_{CS}$ | 65.0 | 65.2 | 91.3 | 79.9 |
| TS ASPIRE$_{Bio}$ | 65.5 | 65.6 | 90.9 | 79.8 |
| OT ASPIRE$_{CS}$ | 64.5 | 65.5 | 91.2 | 79.8 |
| OT ASPIRE$_{Bio}$ | 65.0 | 65.7 | 90.5 | 79.6 |
| SciNCL + MTL CTRL | 66.9 | 66.9 | 91.1 | 80.7 |
| SciNCL + Adapters | 67.0 | 67.5 | 91.5 | 81.1 |

ASPIRE (Mysore et al., 2022) produces representations for the dense retrieval of scientific documents based on matching multiple aspects between the query and candidates. To evaluate these representations under the settings they are designed for, we only report the results on the proximity tasks in Table 7. We use the model implementations available on HuggingFace which have been pre-trained on documents from the Computer Science (CS) and Biomedical (Bio) domains. The models variants can be further sub-categorized as retrieval based on best aspect matching (TS ASPIRE) and a weighted sum of the similarity score among all the aspects based on Optimal Transport (OT ASPIRE) between the query and candidates. Both our multi-format approaches with control codes and adapters produce better results overall and on out-of-train tasks. Note however, since ASPIRE models are trained on co-citations, they perform much better on average on the citation based tasks from SciDocs.

---

[11]We experimented with mixed and task sequential batching as well which did not yield good results.

## D    SCIREPEVAL DOMAIN DISTRIBUTION

Table 8: Data domain distribution in SciRepEval for the training tasks and comparison with SciDocs. We group the unique documents in both the benchmarks by their MAG (Wang et al., 2020a) fields of study and present the counts in columns 2 and 3 and the absolute increase per field in column4.

| Field of study | SciRepEval (A) | SciDocs (B) | Increase Ratio (A/B) |
|---|---|---|---|
| Medicine | 3,201,323 | 74,685 | 43 |
| Computer Science | 1,187,689 | 199,664 | 6 |
| Biology | 882,357 | 13,377 | 66 |
| Chemistry | 508,056 | 3,813 | 133 |
| Psychology | 492,071 | 22,590 | 22 |
| Materials Science | 271,865 | 7,681 | 35 |
| Engineering | 254,826 | 31,444 | 8 |
| Mathematics | 231,482 | 25,800 | 9 |
| Physics | 217,670 | 7,285 | 30 |
| Business | 217,585 | 5,450 | 40 |
| Sociology | 156,128 | 2,305 | 68 |
| Political Science | 154,388 | 1,032 | 150 |
| Economics | 123,357 | 2,705 | 46 |
| Environmental Science | 91,682 | 1,136 | 81 |
| Art | 89,527 | 206 | 435 |
| Geography | 83,688 | 1,491 | 56 |
| Philosophy | 61,996 | 151 | 411 |
| Geology | 51,103 | 640 | 80 |
| History | 46,430 | 159 | 292 |

We study the domain diversity of SciRepEval and display the results in Table 8. To compare against the training data for SciDocs, we consider the citation prediction triplets on which SPECTER is trained which is also a subset of the SciRepEval in-train tasks. Even though Medicine and Computer Science papers still form a bulk of the data, SciRepEval has 105x more documents on average per domain compared to the SPECTER triplets.

## E    SPECTER OBJECTIVE

Lastly, we perform an ablation study to better understand the importance of the unsupervised citation-based training objective. We used SciBERT as the base model for this ablation since both SPECTER and SciNCL were trained with the citation objective. Removing the citation objective and its accompanying data from SciBERT + MTL CTRL, we find that the in-train performance drops from 61.9 to 61.8, while out-of-train drops from 57.9 to 57.5, hinting that the citation objective may be helpful for generalization to new tasks.

## F    CROSS-TASK CORRELATION ANALYSIS

In Figure 2 we show Pearson correlations of model performance metrics between tasks in SciRepEval. To compute the correlations, we include all of the individual task results of the model runs shown in Table 2 and Table 3, excluding the ensembles. The correlations between tasks in SciDocs (bottom right) are highest, while correlations between tasks in the entirety of SciRepEval span a larger range. Notably, DRSM-Complete and S2AND are uncorrelated with most other tasks. This shows that the overall task diversity is larger in SciRepEval than in SciDocs.

## G    TASK RELATEDNESS FOR MULTI-TASK TRAINING

To train on multiple tasks simultaneously, care has to be taken to choose a combination of tasks such that negative transfer is avoided, but finding this optimum combination of tasks is hard (Aribandi

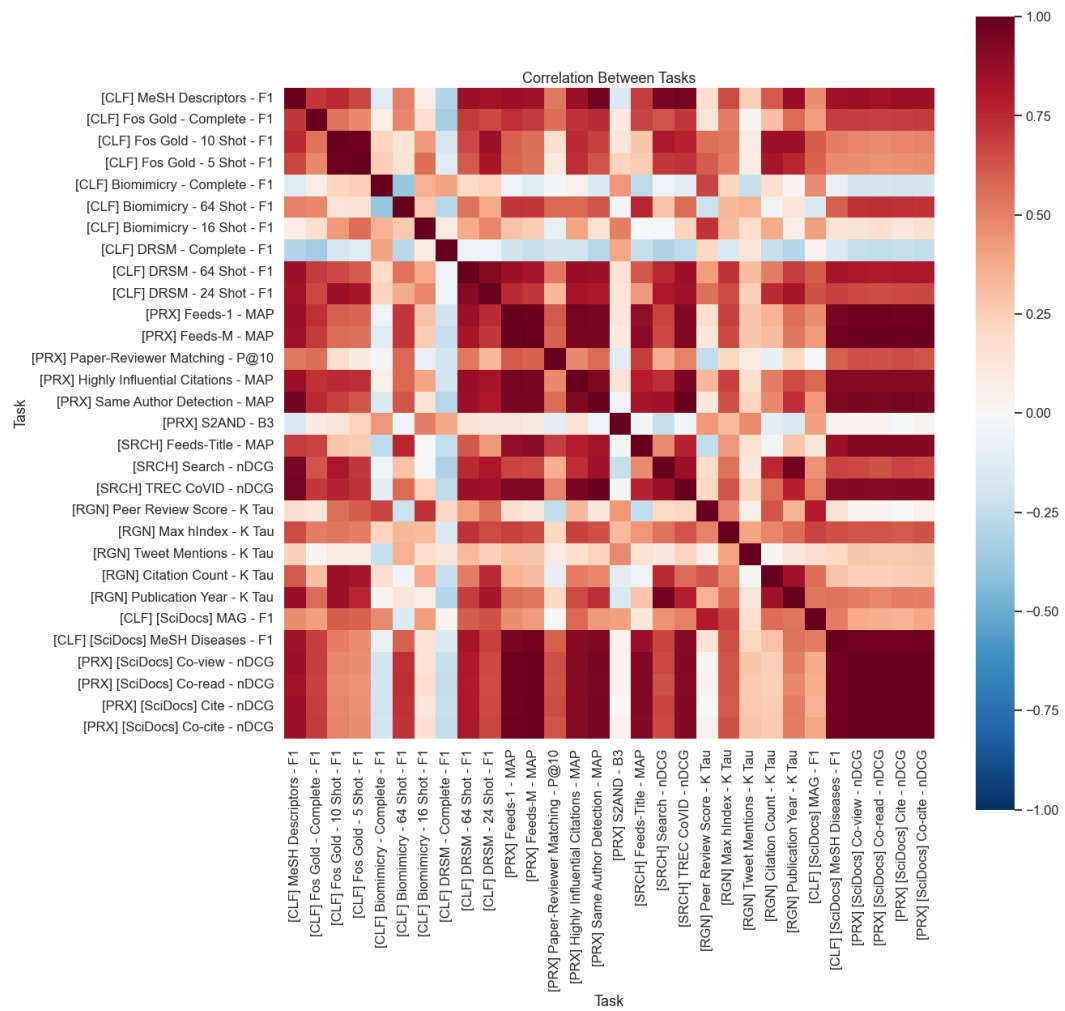

Figure 2: Correlations of model performances between tasks in SciRepEval.

et al., 2022; Padmakumar et al., 2022; Fifty et al., 2021). For a given set of $\mathcal{T}$ tasks, it's often not feasible to train on all $2^{\mathcal{T}} - 1$ task combinations to find the best one. Hence, rather than coming up with an optimum combination, recent work suggests pre-fine-tuning on a large collection of tasks simultaneously to offset the negative transfer between a subset of those tasks(Aghajanyan et al., 2021; Aribandi et al., 2022). Padmakumar et al. (2022) show that pre-fine-tuning on a small set of tasks related to the downstream task is more efficient than large scale multi-task training and yields similar results. As shown in Table 9, we study this training affinity among our task formats by pre-fine-tuning on individual task formats as well their combinations in a multi-task setup. Supporting the findings in Padmakumar et al. (2022), pre-fine-tuning on all the tasks using each individual format's training data leads to better or similar performance on related downstream tasks when compared to training on all the formats at once for all the cases except out-of-train classification. Moreover, combining related task groups like classification and regression during pre-fine-tuning results in better task transfer than training on them individually. This may be due to the fact that the learned representations of both are consumed as features by a downstream linear SVM. Similarly, proximity and ad-hoc search seem to give one another a boost when trained together. However, training on all the tasks simultaneously yields the best results *on average*, and individual results are within 1 point of the best combinations per task format, except for out-of-train regression and in-train classification.

Table 9: Task relatedness analysis for choosing a sub-group of tasks to train on so as to obtain optimum performance. The base SciNCL model(Ostendorff et al., 2022b) is trained on one or more task formats (rows) and then evaluated for a comparison with SciNCL CTRL (last row). Both per task format and overall average performance is reported (columns). The best training combination for every task is highlighted in **bold**. The best single and combined training results for every evaluated task format respectively are underlined.

| Task Format(s) Trained On | | | Out-of-Train | | | | | In-Train | | | All |
|---|---|---|---|---|---|---|---|---|---|---|---|
| | CLF | RGN | PRX | SRCH | Avg | CLF | RGN | PRX | SRCH | Avg | Avg |
| CLF | 56.8 | 15.4 | 70.0 | 77.5 | 54.1 | 68.2 | 21.3 | 59.3 | 71.1 | 52.7 | 64.6 |
| RGN | 56.6 | **21.4** | 68.1 | 74.1 | 54.4 | 53.2 | 45.0 | 55.2 | 68.7 | 53.7 | 59.8 |
| PRX | 55.4 | 19.6 | 73.7 | 84.5 | 57.6 | 57.8 | 32.1 | **67.2** | 73.2 | 55.3 | 68.6 |
| SRCH | 46.6 | 15.7 | 71.7 | 83.8 | 54.1 | 53.2 | 28.1 | 60.4 | 78.0 | 51.6 | 63.3 |
| CLF + RGN | 58.7 | 20.5 | 68.4 | 75.0 | 54.8 | **68.8** | **45.1** | 57.3 | 68.7 | 58.7 | 63.8 |
| CLF + PRX | 57.2 | 16.5 | 73.4 | 82.2 | 56.5 | 68.3 | 32.6 | 66.4 | 43.8 | 58.1 | 68.8 |
| CLF + SRCH | 58.8 | 17.2 | 71.6 | 83.5 | 56.6 | 67.7 | 31.0 | 59.9 | 78.4 | 56.5 | 66.3 |
| RGN + PRX | 51.8 | 20.9 | 73.5 | 78.2 | 56.1 | 50.4 | 44.3 | 67.1 | 71.7 | 56.5 | 67.7 |
| RGN + SRCH | 52.0 | 17.8 | 72.1 | 84.0 | 55.8 | 54.8 | 43.6 | 60.6 | 78.4 | 56.6 | 65.3 |
| PRX + SRCH | 53.2 | 17.8 | **73.8** | **85.1** | 56.8 | 57.9 | 32.1 | 66.6 | 78.2 | 55.9 | 68.3 |
| All | **58.9** | 18.2 | 73.4 | 84.7 | **57.7** | 66.7 | 44.3 | 66.9 | **78.5** | **62.1** | **70.3** |

