# OpenReview forum: "SciRepEval: A Multi-Format Benchmark for Scientific Document Representations"
_ICLR.cc/2023/Conference — Submitted to ICLR 2023_

### Official Review · Reviewer_tJad · 2022-10-22

**Confidence:** 4
**Correctness:** 3
**Technical Novelty And Significance:** 2
**Empirical Novelty And Significance:** 3
**Recommendation:** 6

**Clarity, Quality, Novelty And Reproducibility:**

- The paper is well-written. It does not go into very much technical depth, though it doesn’t need to as a benchmarking paper, although the nature of the data could have been explored more fully. It is an extension of a previous benchmark in this space; while it is a substantial extension, this still limits its novelty.


**Strength And Weaknesses:**

Strengths

- Not only are we treated to a single task in this benchmark, but 25, including classification, regression, and recommendation, each of which have appropriate measures for evaluation (though other measures may also have been appropriate).
- Given the variety of tasks, a ‘multi-format’ method of representation learning is proposed which is not entirely novel on its own, but is additively useful within this paper more ostensibly about the benchmarking suite.
- Suitable baselines (SciBERT, SPECTER) and several extensions (e.g., CTRL-code and embedding adapters) provide a suitable evaluation suite.


Weaknesses

- Although much of the information is found in the Appendices, more detail on the tasks themselves are expected within the main body, especially with regards to the source of the data, which could likely be added to Table 1, in some form.
- Tables 2 and 3 and their discussion in the text require some additional context to evaluate these results, especially with regards to the units of measure and highlighting that the first two columns are not exactly comparable with the third (SciDocs). Given the relative similarities of the performance between SPECTER and SCiNCL (especially considering the variants of each), statistical significance tests should be performed.


Minor

- The use of task-specific control codes is reminiscent (somewhat) of ELMo, and a comparison or acknowledgement should be made to some of the literature around that model.
- Some further explanation around the deficiencies of SciDocs in Sec 2 (especially with regards to the correlations and easiness of negative candidates) really should be provided, especially as SciRepEval’s superiority is meant to be evaluated in contrast to the former.
- There is a higher-than-usual amount of repetitiveness, especially around the 25 tasks in SciRepVal in the first few pages, but this is a very minor complaint
- Some deeper dive into how performance is affected by covariates such as the field of study (Table 8) after one filters out the effect of data set size would be interesting.
- Be sure to check the formatting of your references


**Summary Of The Paper:**

This paper introduces SciRepEval, which is a novel benchmark for training and evaluating scientific document representations. Additionally, the struggles of multi-task learning with regards to generalization in these tasks is explored. An alternative that explicitly encodes the task type is offered and evaluated.


**Summary Of The Review:**

This paper extends an existing benchmark substantially, and offers a new approach to multi-task learning meant to deal with the multitude of tasks in a way that is not wholly novel, and beats the baseline, although only slightly.

---

> ### Author Response · Authors · 2022-11-18
> **Response to reviewer tJad**
>
> Thank you for the thorough and insightful review.  We have addressed the majority of your concerns in our revised version.  We discuss these changes below along with our plans to address each of the remaining concerns.
>
> **Although much of the information is found in the Appendices, more detail on the tasks themselves are expected within the main body, especially with regards to the source of the data, which could likely be added to Table 1, in some form.** \
> We have added a Data Source column to Table 1, also indicating which datasets are new.
>
> **Tables 2 and 3 and their discussion in the text require some additional context to evaluate these results, especially with regards to the units of measure and highlighting that the first two columns are not exactly comparable with the third (SciDocs).** \
> We have now clarified that, since our benchmark includes a variety of tasks with different evaluation metrics, we follow previous benchmarks (e.g. SuperGLUE) in reporting a simple average across tasks.  Further, we now report standard deviations (discussed below) for each column in Table 2  as additional context for interpreting the results.
>
> **Given the relative similarities of the performance between SPECTER and SCiNCL (especially considering the variants of each), statistical significance tests should be performed.** \
> We have rerun the models 4 more times, and Table 2 now shows the mean and standard deviation across all the runs for models we trained. For significance testing we conducted one way analysis of variance (ANOVA) with Tukey’s Range test for post hoc analysis at ɑ=0.05 across the model runs in Table 2 and report the same as well. We find that the difference between the variants is statistically significant. We also find that the overall average results for SciNCL Adapters and SciNCL MTL CTRL + Adapters are not significantly different. We couldn’t perform significance testing for the baseline models (SciBERT/SPECTER/SciNCL) since we use the publicly available model weights for evaluation only, and did not retrain them. Further, we also tested the SciNCL and SPECTER MTL CTRL and found that the differences in their results are not statistically significant (p value of 0.5 for overall average).
>
> **The use of task-specific control codes is reminiscent (somewhat) of ELMo, and a comparison or acknowledgement should be made to some of the literature around that model.** \
> Thank you for the pointer. There is now a brief discussion of ELMo in the related work section.
>
>  We will attempt to **reduce the repetitiveness** you note in our final version.
>
> **Some further explanation around the deficiencies of SciDocs in Sec 2 (especially with regards to the correlations and easiness of negative candidates) really should be provided, especially as SciRepEval’s superiority is meant to be evaluated in contrast to the former.** \
> We have computed the correlation coefficients between all of the task metrics across the entire suite of experiments run for this paper, and the results are in Appendix 5, Figure 2. As can be seen in the figure there, the four nearest neighbors tasks SciDocs are highly self-similar (correlated), while the correlation between non-SciDocs tasks is much more spread out. This shows that the overall task diversity is larger in SciRepEval than in SciDocs.
>
> **Some deeper dive into how performance is affected by covariates such as the field of study (Table 8) after one filters out the effect of data set size would be interesting.** \
> We plan to include an analysis by field-of-study in the camera-ready version.
>
> **Be sure to check the formatting of your references.** \
>  We have fixed some errant references and citations, thank you.
>
> **It is an extension of a previous benchmark in this space; while it is a substantial extension, this still limits its novelty.** \
> We believe our benchmark fills a substantial gap in evaluation of representation learning models for science. We’d also like to reiterate that 11 of our datasets are novel (please see detailed reply to Reviewer #1), and our benchmark allows measuring cross task generalization and multi-task learning, which was not possible with previously existing benchmarks.

---

### Official Review · Reviewer_d6LN · 2022-10-23

**Confidence:** 4
**Correctness:** 4
**Technical Novelty And Significance:** 3
**Empirical Novelty And Significance:** 3
**Recommendation:** 8

**Clarity, Quality, Novelty And Reproducibility:**

Clarity: The paper is written clearly and the logic flows well.

Quality: I believe this is a high quality paper with interesting results and strong justification.

Novelty: The authors identify a challenge with existing benchmarks and release a larger and more diverse benchmark. This is an important and novel contribution.

Reproducibility: Data is promised to be released; no mention of code release.

**Strength And Weaknesses:**

Strengths:
- Introduces a more comprehensive benchmark for scientific document representation learning
- Clearly distinguishes SciRepEval from previous benchmarks (SciDocs) and justifies why a new benchmark is needed (low-power of the recommendation task, a need for more realistic task settings, and lack of training corpus)
- Comprehensive ablation setup targeting fine-grained representation learning, multi-task learning with a strong base model

Weaknesses:
- It would have been nice to see an explicit section detailing what additional challenges remain in benchmarking scientific representation learning and how these limitations apply to the results seen with multi-task approaches.
- Would have liked to see more discussion about task relatedness when discussing benchmarks, especially since SciRepEval seems to be designed in order to improve breadth and coverage of common real-world applications of the representations. For example, it would be interesting to see the relationship as applied to the task of choosing which sub-tasks to group for multi-task learning (see Efficiently Identifying Task Groupings for Multi-Task Learning; Fifty et al. NeurIPS 2021).

**Summary Of The Paper:**

The authors of this paper introduce a benchmark for scientific representation learning consisting of 25 tasks in 4 formats (classificaiton, regression, ranking, search) and evaluate multiple general-purpose scientific representation learning methods (SciBERt, SPECTER, SciNL) alongside adaptation methods to learn task-specific and format-specific representations. The results suggest that fine-grained scientific document representations significantly out-perform general-purpose representations.

**Summary Of The Review:**

The authors identify an area of opportunity relating to current datasets and methods to evaluate scientific representation learning models and propose a well-grounded, larger benchmark in SciRepEval. The ablation study is comprehensive and there is some interesting discussion of cross-task analysis, although I would have liked to see that section expanded.

---

> ### Author Response · Authors · 2022-11-18
> **Response to reviewer d6LN**
>
> We are happy to hear that you liked our paper! We address your questions and concerns below.
>
> **It would have been nice to see an explicit section detailing what additional challenges remain in benchmarking scientific representation learning and how these limitations apply to the results seen with multi-task approaches.** \
> Thank you for pointing out this gap. We have rewritten the conclusion to explicitly state additional challenges: “Future work could address limitations of our work by evaluating partitioning schemes beyond format, crafting higher-fidelity metrics to account for the diversity of tasks in SciRepEval (which vary in sensitivity and relevance to specific applications), or further exploring how accuracy varies with computational and storage cost.”
>
> **Would have liked to see more discussion about task relatedness when discussing benchmarks, especially since SciRepEval seems to be designed in order to improve breadth and coverage of common real-world applications of the representations. For example, it would be interesting to see the relationship as applied to the task of choosing which sub-tasks to group for multi-task learning (see Efficiently Identifying Task Groupings for Multi-Task Learning; Fifty et al. NeurIPS 2021).** \
> Thank you for the suggestion. We expanded our current analysis to study relatedness among the various SciRepEval tasks in our specified task formats for the purpose of multi-task training. We pre-finetune SciNCL on training data from individual task formats as well as their combinations and evaluate on SciRepEval. We find that training on tasks from similar formats such as regression/classification and proximity/ad-hoc search leads to better or similar results on related tasks (from the same formats) when compared to training on all the training tasks in SciRepEval. However, multi-task training on all the tasks gives the best results on average across all the tasks. These results are discussed briefly in Section 7, and fully described in Appendix G (new in the updated draft).
>
> **No mention of code release.** \
> We will be releasing our code, which will facilitate using the SciRepEval datasets for evaluation and training. We have now uploaded an anonymized version of our code for your reference: https://anonymous.4open.science/r/scirepeval-6FDC/

---

### Official Review · Reviewer_RNDk · 2022-11-01

**Confidence:** 3
**Correctness:** 3
**Technical Novelty And Significance:** 1
**Empirical Novelty And Significance:** 2
**Recommendation:** 3

**Clarity, Quality, Novelty And Reproducibility:**

I did not find the paper to be novel, but the quality and clarity of the writing was sufficient

**Strength And Weaknesses:**

Strengths:
- The paper consolidates useful tasks to make one benchmark for scientific document representations

Weaknesses:
- The models mentioned in the paper are mostly created by others in the community
- The benchmark, while useful, is a consolidation of tasks that were already existing


**Summary Of The Paper:**

This paper introduces a new benchmark for scientific document representations, including 25 tasks across classification, regression, ranking and search. The paper also includes investigations on performances of existing models on the introduced tasks.

**Summary Of The Review:**

While the paper provides a useful benchmark, there is a strong component of novelty missing from the paper. ICLR does not seem like the correct venue for this paper to be presented.

---

> ### Author Response · Authors · 2022-11-18
> **Response to reviewer RNDk**
>
> We are glad to hear that you believe our task is useful and the writing is high quality. Here are our replies to your concerns.
>
> **The benchmark, while useful, is a consolidation of tasks that were already existing.** \
> This claim is inaccurate. As we describe in Section 3, the following datasets are new contributions:
> 1. Search - 700k clickthrough events from a scholarly search engine.
> 2. Three feeds evaluation tasks from the scholarly paper recommender.
> 3. Highly influential citations.
> 4. MeSH descriptors.
> 5. Fields of study classification, both silver and gold.
> 6. Predicting citation count.
> 7. Predicting year of publication.
> 8. Peer review score.
> 9. hIndex of authors.
>
> **The models mentioned in the paper are mostly created by others in the community.** \
> The primary objective of this paper is _not_ to explore new methodology for existing datasets, but to provide a novel comprehensive benchmark for training and evaluation of document representation learning methods for scholarly text. That being said, we also provide strong multi-task baseline methods using task-specific adapters and control code methods that outperform existing state-of-the-art.
>
> **ICLR does not seem like the correct venue for this paper to be presented.** \
> Assuming the reviewer thinks our paper does not belong at ICLR because it is a dataset/benchmark paper, we strongly disagree. ICLR has a rich tradition of publishing high-quality large-scale benchmark papers. For example:
> 1. GLUE - https://openreview.net/forum?id=rJ4km2R5t7
> 2. Meta Dataset - large scale benchmark for few shot learning - https://openreview.net/pdf?id=rkgAGAVKPr
> 3. Long Range Arena (LRA) benchmark - https://openreview.net/pdf?id=qVyeW-grC2k
> 4. Gene Disco for Drug Discovery - curated set of existing datasets - https://openreview.net/forum?id=-w2oomO6qgc
> 5. ReClor - a Hard Reading Comprehension benchmark https://openreview.net/forum?id=HJgJtT4tvB
> 6. WILDS 2.0 benchmark for distribution shift https://openreview.net/forum?id=z7p2V6KROOV

---

### Official Review · Reviewer_uueE · 2023-01-06

**Confidence:** 4
**Correctness:** 3
**Technical Novelty And Significance:** 2
**Empirical Novelty And Significance:** 3
**Recommendation:** 6

**Clarity, Quality, Novelty And Reproducibility:**

The paper is well written with high clarity and above-average quality.
Some concerns on the Novelty, since the paper doesn't go deep on the technique.

**Strength And Weaknesses:**

Strength
- The authors provide a comprehensive analysis on 25 tasks in 4 formats.
- The authors use strong baseline models to make the results more convincing.

Weakness
- There is not enough information on the tasks in the main content. Would be better to provide some high-level info there and left the majority in the Appendix.

**Summary Of The Paper:**

This paper introduces a new benchmark (SciRepEval) for scientific representation learning consisting of 25 tasks in 4 formats (classification, regression, ranking, and search). It shows that learning a separate document representation for each task format would improve the task performance compared to learning a single representation for all tasks.

**Summary Of The Review:**

This paper is written in above-average quality and provides a new benchmark with a comprehensive analysis

---

### Decision · Program_Chairs · 2023-01-20

**Decision:**

Reject

**Justification For Why Not Higher Score:**

The paper presents a new benchmark and is well-written. Besides introducing additional tasks in the new dataset, there is not much novel contribution.


**Justification For Why Not Lower Score:**

The new benchmark is important for improving scientific document representation and modelling on the most commonly used tasks.

**Metareview: Summary, Strengths And Weaknesses:**

The paper presents a new benchmark, SciRepEval, for evaluating scientific document representation models. SciRepEval contains 11 additional tasks compared to the benchmark SciDocs. The paper also argues the necessity of developing the new benchmark and supports its arguments by demonstrating the performance loss of the SOTA scientific document representation models on SciRepEval compared to the SciDocs benchmark. The authors also did a good job addressing the reviewers' comments in the rebuttal. However,  besides introducing additional tasks in the new dataset, there is not much novel contribution.